# Modeling susceptibility to drug-induced long QT with a panel of subject-specific induced pluripotent stem cells

Francesca Stillitano[1†], Jens Hansen[2†], Chi-Wing Kong[1], Ioannis Karakikes[1], Christian Funck-Brentano[3], Lin Geng[1], Stuart Scott[4], Stephan Reynier[5], Ma Wu[5], Yannick Valogne[5], Carole Desseaux[5], Joe-Elie Salem[3], Dorota Jeziorowska[3], Noël Zahr[3], Ronald Li[1,6,7], Ravi Iyengar[2], Roger J Hajjar[1], Jean-Sébastien Hulot[1,3]*

[1]Cardiovascular Research Center, Icahn School of Medicine at Mount Sinai, New York, United States; [2]Department of Pharmacology and Systems Therapeutics, Systems Biology Center, Icahn School of Medicine at Mount Sinai, New York, United States; [3]Sorbonne Universités, UPMC Univ Paris 06, AP-HP, INSERM, CIC-1421, Institute of Cardiometabolism and Nutrition, Paris, France; [4]Department of Genetics and Genomic Sciences, Icahn School of Medicine at Mount Sinai, New York, United States; [5]Cellectis Stem Cells, Paris, France; [6]Ming Wai Lau Centre for Reparative Medicine, Karolinska Institutet, Stockholm, Sweden; [7]Dr. Li Dak-Sum Centre, The University of Hong Kong – Karolinska Institutet Collaboration in Regenerative Medicine, Pokfulam, Hong Kong

*For correspondence: jean-sebastien.hulot@aphp.fr

[†]These authors contributed equally to this work

**Abstract** A large number of drugs can induce prolongation of cardiac repolarization and life-threatening cardiac arrhythmias. The prediction of this side effect is however challenging as it usually develops in some genetically predisposed individuals with normal cardiac repolarization at baseline. Here, we describe a platform based on a genetically diverse panel of induced pluripotent stem cells (iPSCs) that reproduces susceptibility to develop a cardiotoxic drug response. We generated iPSC-derived cardiomyocytes from patients presenting in vivo with extremely low or high changes in cardiac repolarization in response to a pharmacological challenge with sotalol. In vitro, the responses to sotalol were highly variable but strongly correlated to the inter-individual differences observed in vivo. Transcriptomic profiling identified dysregulation of genes (*DLG2, KCNE4, PTRF, HTR2C, CAMKV*) involved in downstream regulation of cardiac repolarization machinery as underlying high sensitivity to sotalol. Our findings offer novel insights for the development of iPSC-based screening assays for testing individual drug reactions.

## Introduction

A large number of drugs have the undesirable side effect of prolonging cardiac repolarization which can trigger life-threatening cardiac arrhythmias (*Kannankeril et al., 2010*). A typical cause is the inhibition of the inward rectifying potassium channel (hERG or *human ether-a-go-go* related gene encoded by *KCNH2*) (*Kannankeril et al., 2010*; *Roden, 2008a*), which results in the prolongation of the repolarization time and induces an increase of the QT interval on the electrocardiogram. The early prediction of drug-induced long QT (diLQT) is a major requirement to avoid exposing patients to cardiac arrhythmias and ultimately sudden cardiac death (*Sarganas et al., 2014*). Eventually, diLQT is a major cause of drug withdrawal or of premature termination during development (*Fermini and Fossa, 2003*). Regulatory agencies require that new drug candidates be systematically

**eLife digest** Common medications can disturb the electrical signals that cause the heart to beat, potentially resulting in sudden death. Many of the drugs that have these "cardiotoxic" effects were not designed to affect the heart, and include anti-allergenics and anti-vomiting drugs. In general, only a small proportion of individuals treated with these drugs will be at risk of fatal side effects; this risk variation is thought to be due to genetic differences. If these people could be reliably identified, the drugs could be used to treat others who will not develop cardiotoxic reactions, but it is difficult to predict the effect a drug will have on the beating of the heart.

Stillitano, Hansen et al. have now investigated whether skin cells can be used to predict an individual's likelihood of developing cardiotoxic side effects. Skin cells can be reprogrammed to form pluripotent stem cells, which have the ability to develop into any of the cell types in the adult body – including heart muscle cells. The effects of drugs could then be tested on these artificially created heart cells, yet it is not clear whether these effects would be the same as those seen in actual heart cells

Stillitano, Hansen et al. created heart cells from skin samples collected from many different people and treated the cells with a drug that affects the rhythm of the heart. Some of the cells came from people whose heart rhythm is strongly affected by the drug, and others came from people whose heart rhythm is barely altered. The response of the lab-grown cells was closely related to whether the cells came from a person who was susceptible to the effects of the drug. Further investigation revealed that the genes that are important for maintaining a regular heartbeat differ in people who experience strong cardiotoxic side effects from those that do not.

Overall, the results presented by Stillitano, Hansen et al. support the idea that induced pluripotent stem cells could be used to predict an individual's risk of developing cardiotoxic reactions. Further work is now needed to develop this approach.

screened for hERG inhibition in animal models or in heterologous over-expression cellular systems (*Sager et al., 2014*). Because of species-dependent differences of cardiac electrophysiology and the failure of heterologous systems to express all cardiac proteins, hERG screening however remains suboptimal and has a poor positive predictive value for subsequent proarrhythmia risk (*Gintant, 2011*; *Giorgi et al., 2010*; *Sager et al., 2014*).

The risk of developing diLQT varies markedly between subjects. Previous studies in the general population have shown that a small proportion (i.e., less than 10%) of patients exposed to QT-prolonging drugs actually develop diLQT (*Behr and Roden, 2013*; *Kannankeril et al., 2010*; *Soyka et al., 1990*). Different observations support the existence of a genetic determinism to diLQT including the higher propensity to develop drug-induced repolarization abnormalities in first-degree relatives of patients with diLQT (*Kannankeril et al., 2005*) and the similarity to the congenital form of the long QT syndrome (LQTS), a rare inherited cardiac disorder (*Moss and Robinson, 1992*). Multiple rare mutations with marked effects in genes encoding ion channels have been reported to underlie LQTS (*Cerrone and Priori, 2011*). However, candidate gene screening studies in large cohorts of patients with diLQT have reported a small prevalence (from 10 to 15%) of these mutations in the known LQTS genes (*Paulussen et al., 2004*; *Yang et al., 2002*). QT prolongation can be induced by mutations in many genes that occupy a substantial region of the human interactome and contain many drug targets (*Berger et al., 2010*). The factors leading to diLQT in some individual subjects remain largely unclear but are probably driven by the additive effect of common genetic variants that reduce the cardiac repolarization reserve (*Behr and Roden, 2013*; *Kannankeril et al., 2005*; *Roden, 2006*, *2008b*). As a consequence, diLQT is difficult to predict as it usually develops in individuals with a predisposing genetic makeup which favors exaggerated response to a pharmacological challenge with QT-prolonging drugs (*Kannankeril et al., 2010*; *Roden, 2008b*) but does not affect cardiac repolarization features at baseline (*Behr and Roden, 2013*; *Kannankeril et al., 2010*; *Sarganas et al., 2014*).

Human induced pluripotent stem cells (iPSCs) have recently been proposed to model monogenic disorders including congenital LQTS (*Itzhaki et al., 2011*; *Moretti et al., 2010*; *Sinnecker et al.,*

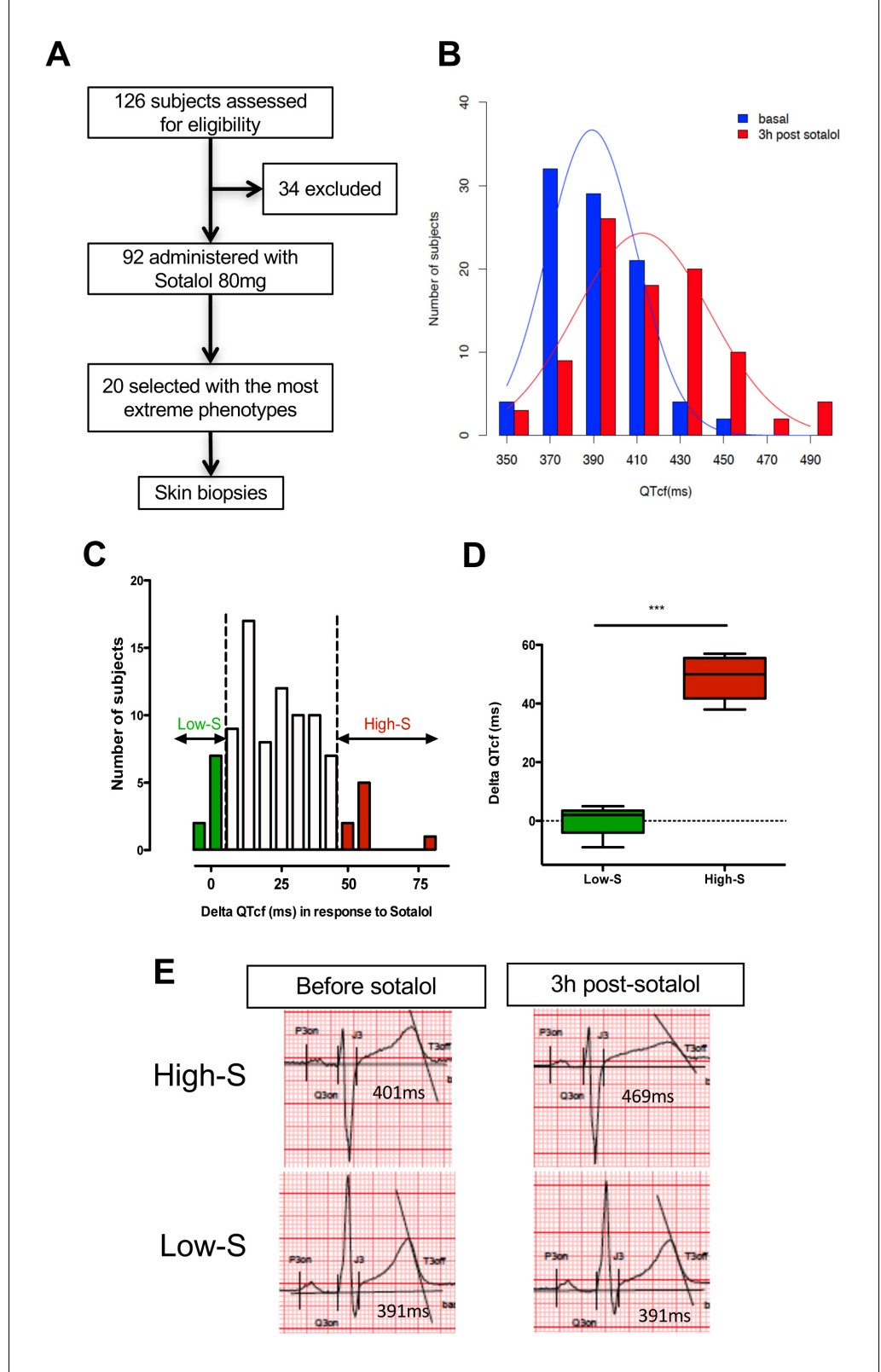

**Figure 1.** QTcf changes following Sotalol administration in healthy volunteers. (**A**) Flow chart of the clinical study. (**B**) Distribution of QTcf duration before (blue) and 3 hr after sotalol intake (red). (**C**) Distribution of delta change in QTcf showing the wide inter-individual variability in response to the same pharmacological stimulation. Subjects with the most extreme responses were selected as low sensitive (low-S) or high- sensitive (high-S) as indicated in

*Figure 1 continued on next page*

*Figure 1 continued*

green and red respectively. (**D**) Average delta change in QTcf in the two groups of selected subjects. \*\*\*p<0.001. (**E**) Typical ECG recordings before and after sotalol intake in a high-S subject (upper panels) and in a low-S subject (lower panels). There is one figure supplement.

The following figure supplement is available for figure 1:

**Figure supplement 1.** Overview of the clinical and experimental study and role of each partner.

*2013*). Cardiomyocytes derived from congenital LQTS patient-specific iPSCs lines typically exhibit significant electrophysiological abnormalities due to the deficient function of some key cardiac ion channels (*Itzhaki et al., 2011*; *Moretti et al., 2010*). These cells display increased arrhythmogenicity under native conditions and can be used as a test system for customization of anti-arrhythmic drugs in these patients (*Terrenoire et al., 2013*; *Wang et al., 2014*). In addition, QT prolonging drugs can trigger changes in the electrical potential developed by iPSC-derived cardiomyocytes (iPSC-CMs) (*Liang et al., 2013*; *Navarrete et al., 2013*; *Nozaki et al., 2014*).

To expand upon these observations, our goal was to determine whether iPSC-CMs from a group of subjects with quantitative measure of diLQT in vivo could recapitulate the phenotype in vitro thereby providing a model system for diLQT. In this study, we show that iPSC-CMs from subjects shown to be susceptible to diLQT in vivo but otherwise normal present no abnormalities in their basal characteristics but higher cardiotoxic responses to drug stimulation as compared to control iPSC-CMs. The contrasted responses to sotalol in iPSC-CMs were strongly correlated to the inter-individual differences in response to sotalol as recorded during clinical investigations. Our findings offer novel insights on the use of a genetically diverse panel of induced pluripotent stem cells to model complex pharmacogenetic traits.

## Results

### Identification of subjects susceptible to develop drug-induced long QT

We first performed a clinical study (clinicaltrials.gov NCT01338441) where we prospectively evaluated cardiac repolarization of healthy subjects in response to a pharmacological challenge with a single 80 mg oral dose of sotalol. Sotalol is a non-selective competitive $\beta$-adrenergic receptor blocker with additional Class III antiarrhythmic properties by its inhibition of potassium channels (*Zanetti, 1993*). Sotalol is widely recognized as a classical QT-prolonging drug (*Soyka et al., 1990*). A total of ninety-two subjects were enrolled in the study (*Figure 1A*). All subjects gave their written informed consent to participate the study. Included subjects were aged from 18 to 40 years (30.1 ± 6.2 years), had a body mass index (BMI) between 19 and 29 kg/m$^2$ (23.9 ± 2.3 kg/m$^2$), were

**Table 1.** Related to *Figure 1*. Demographic, clinical and electrocardiographic baseline characteristics of subjects with low- vs. high- sensitivity to Sotalol.

|  | Low-sensitivity, n = 10 | High- sensitivity, n = 10 | p-value |
|---|---|---|---|
| Age | 29.9 ± 5.9 | 28.3 ± 5.9 | 0.53 |
| Gender, Male (%) | 8 (80.0%) | 1 (10%) | 0.006 |
| Body Mass Index (kg/m$^2$) | 23.6 ± 0.9 | 22.9 ± 2.7 | 0.40 |
| SBP (mmHg) | 116.0 ± 8.6 | 113.7 ± 8.3 | 0.50 |
| DBP (mmHg) | 72.2 ± 4.8 | 68.0 ± 4.2 | 0.07 |
| Resting heart Rate (bpm) | 59.6 ± 5.9 | 63.2 ± 8.4 | 0.37 |
| PR interval (ms) | 166.7 ± 24.7 | 143.5 ± 18.1 | 0.09 |
| QRS (ms) | 88.3 ± 5.9 | 85.6 ± 7.7 | 0.08 |
| QTcf (ms) | 384.7 ± 26.4 | 402.6 ± 20.8 | 0.13 |

from Caucasian origin and had no known significant disease or long-term treatment. Forty-four percent (n = 40) were male. Cardiac repolarization parameters were analyzed through standardized measurements of digital high-resolution ECG (sampling rate 1000 Hz). All subjects had a normal sinus rhythm and no significant conduction or repolarization abnormalities. Basal QT interval ranged from 340 to 458 ms (393.4 ± 26.3 ms), with a corresponding heart rate ranging from 47 to 88 bpm (63.3 ± 8.7 bpm). QT interval was corrected according to Fridericia's formula ($QTcf=QT/[RR]^{1/3}$) providing corrected QTcf in the physiological range from 350 to 447 ms (399.0 ± 18.6 ms). Three hours after sotalol intake, QTcf had increased by 23.4 ± 2.4 ms. A large inter-individual variability was however noticed with a change in QTcf ranging from −9 ms to +81 ms and a resulting post-sotalol QTcf ranging from 351 to 489 ms (412.6 ± 43.0 ms) (*Figure 1B*).

Twenty subjects with the most extreme responses to sotalol (i.e., 10 subjects with high-sensitivity (high-S) and 10 subjects with low-sensitivity (low-S)) underwent skin biopsy (*Figure 1A and C*). These subjects displayed highly contrasted responses to the same pharmacological stimulation (average change in QTcf: 48.5 ± 7.1 ms in high-S subjects vs. 0.2 ± 4.8 ms in low-S subjects; p<0.0002; *Figure 1C–E*) while having no significant differences in their basal ECG characteristics (*Table 1*). Demographic characteristics were well balanced between groups with the exception of gender, subjects with high-S being mostly female (90%, n = 9) whereas subjects with low-S were mostly males (80%, n = 8) (*Table 1*). Sotalol plasma level was measured 3 hr after sotalol intake and was not significantly different in subjects with high-S as compared to the other participating subjects (753.1 ± 210.9 vs. 620.7 ± 195.3 µg/ml, p=0.10).

## Derivation and differentiation of iPSCs into cardiomyocytes

Dermal fibroblasts were collected from the twenty subjects with high or low-sensitivity to sotalol stimulation. Human iPSCs were derived through retroviral infection of dermal fibroblasts with the reprogramming factors OCT4, SOX2, c-MYC and KLF4 with a successful generation for 17 out of the 20 subjects (*Figure 1—figure supplement 1*). All iPSC clones expressed the characteristic human

**Table 2.** Related to *Figure 2*. Fibroblasts and iPS quality control parameters.

| ID | Mycoplasma* | HIV*, HBV*, HCV*, HTLV1 and 2* | Phosphatase alkaline staining | Karyotyping |
|---|---|---|---|---|
| P11007~5924~iPSpolyRoksmA | Negative | Negative | Positive | 46,XY |
| P11008~5444~iPSpolyRoksmB | Negative | Negative | Positive | 46, XX |
| P11009~6426~iPSpolyRoksmC | Negative | Negative | Positive | 46, XY |
| P11013~5744~iPSpolyRoksmD | Negative | Negative | Positive | 46, XX |
| P11014-5864-iPSpolyRoksmC | Negative | Negative | Positive | 46, XX |
| P11015~6345~iPSpolyRoksmE | Negative | Negative | Positive | 46, XX |
| P11018~5644~iPSpolyRoksmB | Negative | Negative | Positive | 46, XX |
| P11019-6444-iPSpolyRoksmB | Negative | Negative | Positive | 46, XY |
| P11020 ~ 7125 ~ iPSpolyRoksmD | Negative | Negative | Positive | 46, XY |
| P11021 ~ 6544 ~ iPSpolyRoksmC | Negative | Negative | Positive | 46, XX |
| P11023~5525~iPSpolyRoksmA | Negative | Negative | Positive | 46, XX |
| P11024~5844~iPSpolyRoksmA | Negative | Negative | Positive | 46, XX |
| P11026~6504~iPSpolyRoksmD | Negative | Negative | Positive | 46, XY |
| P11028~6904~iPSpolyRoksmJ | Negative | Negative | Positive | 46, XY |
| P11029-6284-iPSpolyRoksmB | Negative | Negative | Positive | 46, XX |
| P11030~5684~iPSpolyRoksmA | Negative | Negative | Positive | 46, XX, t(1;16)[†] |
| P11031~5204~iPSpolyRoksmC | Negative | Negative | Positive | 46, XY |

*Tested in originating fibroblast cell lines.
[†]As found in originating fibroblasts.

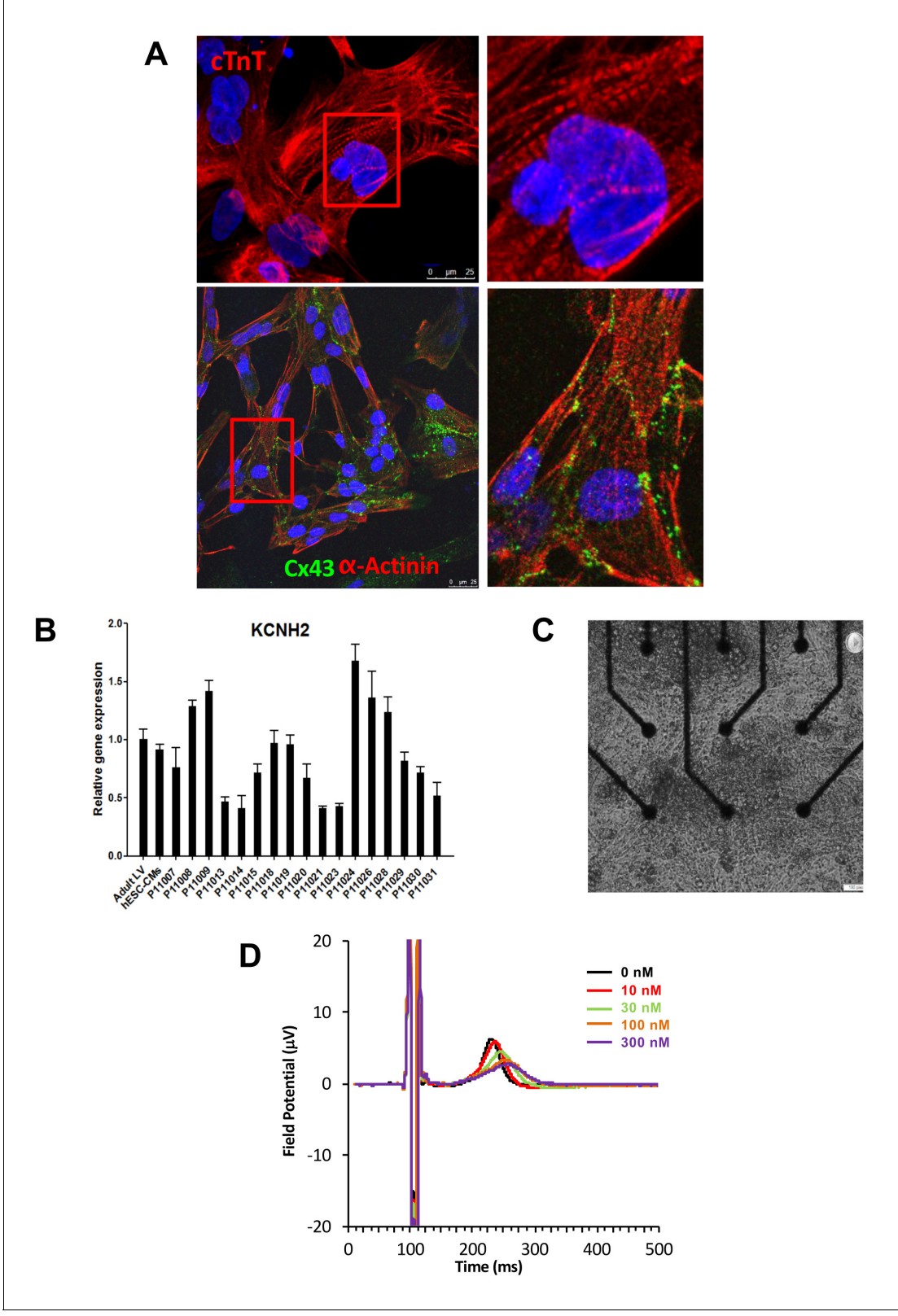

**Figure 2.** Expression of sarcomeric proteins and ion channels in human iPSC-CMs. (**A**) Confocal microscopy imaging of Troponin T (top), alpha-actinin and connexin 43 (bottom) in single generated iPSC-CMs (from line P11015). Nuclei are stained with DAPI (Blue). (**B**) Gene expression of cardiac ion channel *KCNH2* (encoding hERG) by quantitative PCR; Adult LV tissue is used as a positive control and the level of expression in human ESC-derived cardiomyocytes as a comparator. (**C**) Example of monolayer of iPSC-CMs seeded and attached on a 6-well MEA chip, each well containing nine

*Figure 2 continued on next page*

*Figure 2 continued*

microelectrodes (black). See also *Video 1*. (D) Representative field potential duration (FPD) recorded before and after application of the hERG blocker E4031 (from line P11007). There are five figure supplements.

The following source data and figure supplements are available for figure 2:

**Source data 1.** iPSCs characterization.

**Source data 2.** Expression of main human cardiac ion channels.

**Figure supplement 1.** Expression of exogenous and endogenous pluripotency genes.

**Figure supplement 2.** MEA arrays.

**Figure supplement 3.** Quality of FPD adjustment.

embryonic stem cell (ESC) pluripotency markers TRA-1–60, NANOG, OCT3/4 and SSEA-4 (*Figure 2—source data 1*), had positive alkaline phosphatase staining. Sixteen iPSC clones had a normal karyotype (*Table 2*). One clone presented a significant translocation that was observed in originating fibroblasts. All clones showed silencing of the exogenous retroviral transgenes and reactivation of the endogenous pluripotency genes OCT3/4, NANOG and SOX2 (*Figure 2—figure supplement 1*).

The generated iPSCs lines were then encoded to ensure that all further biological experiments were performed in a blinded fashion without knowledge of the associated clinical phenotype (*Figure 1—figure supplement 1*). The code was kept securely by a blinded third party and was only disclosed once all characterizations and electrophysiological measurements with microelectrode array (MEA) mapping system were finalized. All classifications of lines according to the observed response to sotalol were therefore performed before the unblinding process.

Subject-specific iPSCs were differentiated using a small molecule-mediated directed differentiation protocol that involves sequential modulation of the canonical Wnt pathway and yields a high proportion of ventricular-like cardiomyocytes as previously described (*Karakikes et al., 2014*). As early as seven days following initiation of cardiac differentiation, spontaneously beating embryoid bodies (EBs) appeared. To test for iPSC-derived cardiomyocytes quality and purity we first performed immunocytostaining studies showing positive staining for the sarcomeric proteins troponin T (cTnT), and alpha-actinin and for the gap-junction protein connexin 43 (Cx43) (*Figure 2A*). Cardiac troponin T staining of iPSC-CMs showed a typical striated pattern (*Figure 2A*, upper panels). We also determined the cardiomyocytes differentiation efficiency by flow cytometry and found an average of 47.8 ± 19.9% cTNT positive cells. Importantly, after unblinding the study, we did not observe any differences between iPSC lines from the high-S vs. low-S groups (49.2 ± 24.4% vs. 46.7 ± 17.3% respectively, p=0.42). Finally, in line with previous reports (*Liang et al., 2013*), quantitative PCR demonstrated the presence of major cardiac ion channel genes found in adult left ventricular tissue, including of the relevant *KCNH2* gene (*Figure 2B* and *Figure 2—source data 2*).

## Genetic profiling of cell lines

We then screened for common single nucleotide polymorphisms (SNPs, total of 4130) in the 15 genes associated with congenital long QT (*AKAP9, ANK2, CACNA1C, CALM1, CALM2, CAV3, KCNH2, KCNE1, KCNE2, KCNJ2, KCNJ5, KCNQ1, SCN4B, SCN5A, SNTA1)* using a high-density chip (Human Omni 2.5 genotyping array). We found 25 SNPs in *ANK2, SCN5A, KCNQ1, CACNA1C, CALM1, KCNE2, KCNH2* and *KCNJ5* as being significantly imbalanced between low-S vs. high-S groups (*Table 3*). This suggests the presence of an allelic series comprising multiple variants of unknown significance that can create a particular predisposing genetic background. Of note, the *CALM1* c.*1952C (rs3814843) mutated allele was recently associated with increased risk of sudden cardiac death in patients with heart failure (*Liu et al., 2015*). The allele is rare in Europeans (anticipated allelic frequency of 1.8%) but displayed a 22.2% allelic frequency in the high-S group. In addition, the rare C allele of *KCNH2*:c.307 + 1932G>C polymorphism (rs3778873) was significantly more frequent in high-S as compared to low-S (minor allelic frequency: 44.4% vs. 6.2% respectively,

**Table 3.** Related to *Figure 2*. Single nucleotide polymorphisms in *ANK2, SCN5A, KCNQ1, CACNA1C, CALM1, KCNE2, KCNH2* and *KCNJ5* as being significantly imbalanced (p<0.05) between low-S vs. high-S groups. Anticipated minor allelic frequency (MAF) was defined using the HapMap-CEU European data. The MAF in cells from the high-S vs. low-S groups was determined once the study was unblinded. Fisher's exact test was used to compare observed MAFs in high-S vs. low-S groups.

| Gene | Rs number | Anticipated MAF | Observed MAF Low-S | Observed MAF High-S | p value |
|---|---|---|---|---|---|
| ANK2 | rs17045935 | 6.8% | 0% | 22.2% | 0.03 |
| | rs62314901 | 40% | 56.25% | 27.8% | 0.037 |
| | rs17676256 | 10.2% | 12.5% | 38.9% | 0.03 |
| | rs967099 | 38.1% | 25% | 44.4% | 0.027 |
| | rs4834321 | 46.9% | 37.5% | 55.6% | 0.02 |
| | rs35308370 | 48.3% | 25% | 66.7% | 0.025 |
| | rs931838 | 42.5% | 31.25% | 66.7% | 0.048 |
| SCN5A | rs7375123 | 19.2% | 43.75% | 11.2% | 0.007 |
| | rs12491987 | 7.6% | 0% | 22.2% | 0.03 |
| | rs9871385 | 34.1% | 6.25% | 33.3% | 0.02 |
| | rs9818148 | 19.2% | 56.25% | 16.7% | 0.049 |
| KCNQ1 | rs4255520 | 15.5% | 18.75% | 0% | 0.043 |
| | rs151288 | 32.7% | 0% | 22.2% | 0.031 |
| | rs718579 rs11022996 | 39.8% 41.2% | 12.5% | 27.8% | 0.034 |
| | rs151212 | 40.8% | 50% | 33.3% | 0.05 |
| CACNA1C | rs3794299 | 12.4% | 18.75% | 0% | 0.043 |
| | rs4765661 rs2238018 | 15.8% 17.3% | 31.25% | 5.6% | 0.027 |
| CALM1 | rs3814843 | 1.8% | 0% | 22.2% | 0.031 |
| | rs2300502 | 7.5% | 6.25% | 33.3% | 0.023 |
| KCNE2 | rs28409368 | 30.8% | 18.75% | 38.9% | 0.0239 |
| KCNH2 | rs2072411 | 37.5% | 18.75% | 44.4% | 0.027 |
| | rs3778873 | 15.8% | 6.2% | 44.4% | 0.02 |
| KCNJ5 | rs7924416 | 24.6% | 0% | 27.8% | 0.013 |

p=0.02). *KCNH2* encodes for the hERG potassium channel that is targeted by sotalol and this variant was identified in a large genome-wide association study on the physiological regulation of QT interval (*Pfeufer et al., 2009*).

## The library of iPSC-CMs reproduces susceptibility to develop cardiotoxic drug response

To evaluate the electrophysiological properties of iPSC-CMs, we used a microelectrode array (MEA) mapping system (*Figure 2—figure supplement 2*). iPSC-CMs were seeded onto a 6-well MEA chamber in order to form a monolayer in contact with the electrodes. Few days after seeding iPSC-CMs regain spontaneous beating activity thus allowing recordings of field potential duration (FPD)

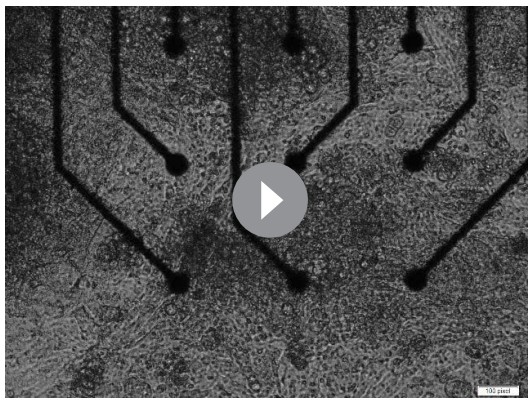

**Video 1.** Spontaneous beating activity of monolayer iPSC-CMs seeded onto one well of a 6-well MEA chamber.

(*Figure 2C–D* and *Video 1*). FPD is analogous to the QT interval in the electrocardiogram (ECG) and was previously shown to correlate with action potential duration (APD) and can be used to test drug effects on repolarization (*Navarrete et al., 2013*). IPSC-CMs were first tested for their responses to E4031, a specific and potent experimental hERG blocker. E4031 resulted in a dose-dependent increase in FPD with typical flattening of the waveform in all but two cell lines (*Figure 2D* and *Figure 2—figure supplements 2* and *3*). There was moderate inter-line variability, except for line P11019 which demonstrated higher sensitivity to E-4031 and was found to belong to the high-S group after unblinding. There was however no significant differences in sensitivity to E4031 between groups as assessed by computations of half maximal effective concentration EC50 ($2.4 \pm 0.7 \times 10^{-8}$ vs $6.6 \pm 3.0 \times 10^{-8}$ M in high-S and low-S groups respectively, p=0.59) or of maximal effect Emax ($49 \pm 20\%$ vs $30 \pm 2\%$ in high-S and low-S groups respectively, p=0.76). Two lines displayed no responses to E4031 and were found as having the lowest expression of hERG (P11021 and P11023, *Figure 2B*). Because of the lack of response to a pure hERG blocker and of appropriate recordings to detect effects of QT prolonging drugs with the MEA system, sotalol response was not further tested on these two lines.

The response to the clinically relevant drug sotalol was then tested in a blinded fashion in the remaining lines. For quality purposes, results from the iPSC line presenting a significant translocation were excluded from further analyses. However, the inclusion of the results obtained with this iPSC line did not change the statistical significance of the results. Final analyses were thus performed on a total of 14 lines (7 vs. 7 lines in low-S and high-S groups respectively).

After unblinding, we found that the different recordings showed a significantly higher response to sotalol in the CMs generated from iPSCs derived from high-S subjects as compared to low-S subjects (*Figure 3A–D*). Increasing sotalol concentrations was associated with a significant prolongation in FPDs in all iPSC-CMs, an effect that was however significantly enhanced in lines from the high-S group (*Figure 3A*). There was a dose-dependent sotalol-induced FPD prolongation that typically appears for sotalol concentrations above 30 µmol/L (*Figure 3A*), as seen in other reports (*Navarrete et al., 2013*). Of note, the application of low sotalol concentration (10 µmol/L), a concentration this is closer to typical sotalol plasma levels, was already associated with a significant increase in FPD in lines from the high-S group as compared to low-S (p=0.03). The maximal FPD prolongation observed in response to sotalol was also significantly higher in the high-S as compared to low-S iPSC-CMs ($52 \pm 6\%$ vs. $27 \pm 6\%$, p=0.02, *Figure 3B*). Sotalol is known to induce arrhythmic events including ectopic beats and short-long-short rhythm. We found that 5 of the seven lines derived from the high-S group displayed arrhythmias in response to sotalol stimulation as they showed development of irregular spontaneous beating rate. On the other hand, arrhythmias only occurred in one of the eight lines from the low-S group (*Figure 3C*). We plotted the relative change in FPD in response to sotalol 30 µM against the observed change in QTcf in donors (*Figure 3D*). With the exception of one cell line in each group, the remaining six low-S and six high-S cell lines were correctly discriminated according to the clinical phenotype. These data indicate that the contrasted responses to sotalol in iPSC-derived CMs are strongly related to the inter-individual differences in response to sotalol as recorded during clinical investigations.

We also evaluated the effect of sotalol on action potential duration (APD) measured in a representative iPSC-CMs line from each group (P11009 as low-S and P11029 as high-S). Experimenter for APD measurement was however kept blinded of the observed drug sensitivity of selected iPS lines. Sotalol (100 and 300 µM) significantly prolonged the action potential, the duration of 90% repolarization ($APD_{90}$) of high-S (P11029) iPSC-CMs ($289 \pm 22$ in drug-free vs. $492 \pm 66$ and $435 \pm 92$ after

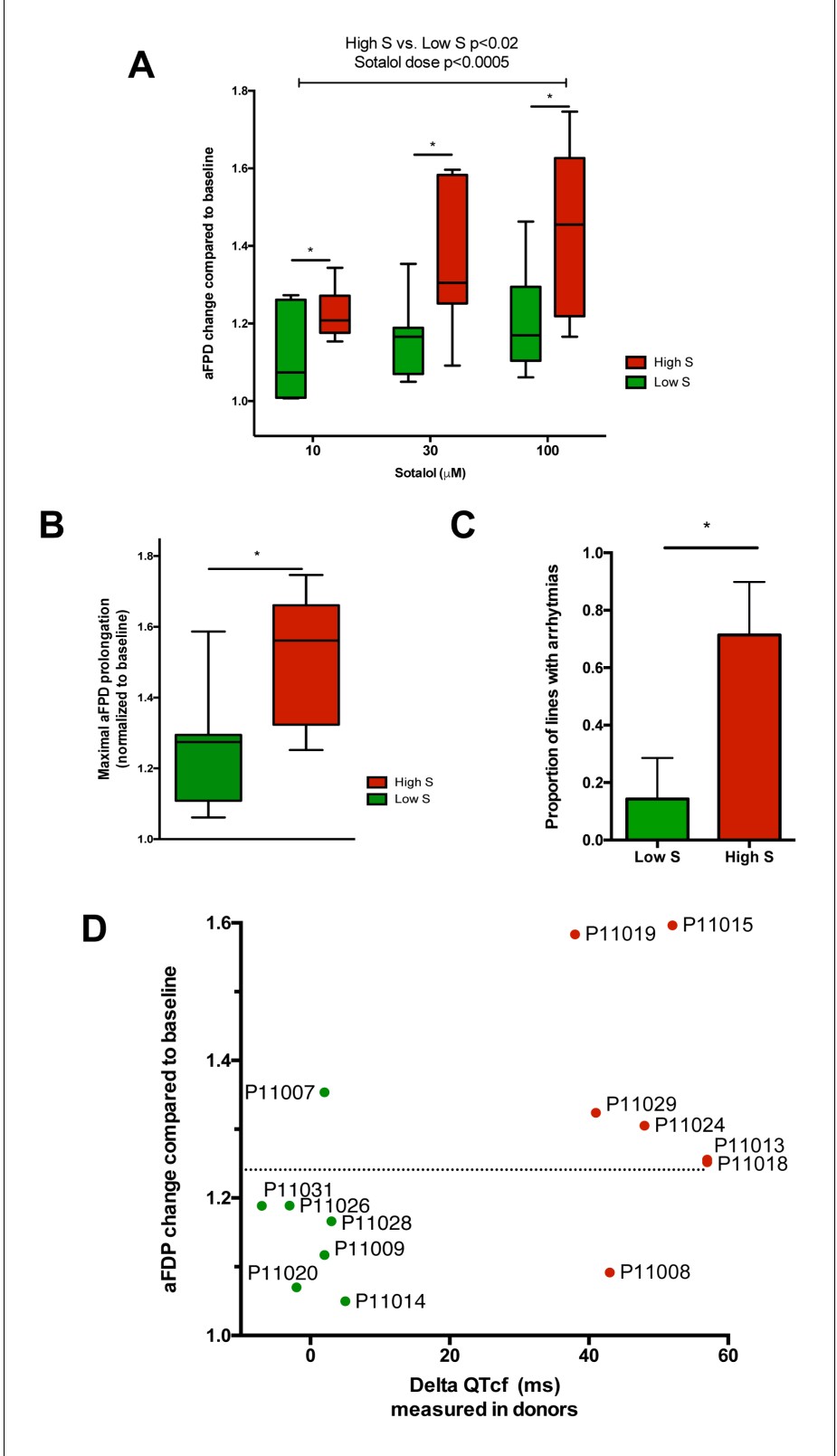

**Figure 3.** Differences in iPSC-CMs responses to Sotalol stimulation according to clinical sensitivity to Sotalol. (**A**) Adjusted FPD (aFPD) measured in iPSC-CMs derived from subjects with low-sensitivity (green) vs. high-sensitivity (red) in response to increasing concentrations of sotalol. aFPD are normalized to baseline values to account for inter-lines variability in aFPD values. Two-way analysis of variance demonstrates a significant influence of sotalol concentrations (p<0.0005) and of the sensitivity group (p<0.02). *p<0.05 for post-hoc comparison between groups; high-S vs. low-S. N = 2–5

*Figure 3 continued on next page*

*Figure 3 continued*

recordings per cell lines per concentrations. (B) Maximal change in aFPD observed during sotalol stimulation. (C) Proportion of observed arrhythmias after sotalol application. (D) Data plot graph showing the correlation between aFPD observed in iPSC-CMs and the DeltaQTcf observed in donors. The aFPD data are reported for sotalol 30 µM concentration. Data points are clustered in two distinct groups. Except for two lines (one line in each group), a threshold a 25% in aFPD change (dashed line) correctly discriminates cells from both groups.

100 and 300 µM sotalol, respectively n = 5) but not that of the low-S line (P11009) even at high dose (339 ± 63 vs. 450 ± 77 ms after 300 µM sotalol, n = 5) (*Figure 4A–B* and *Table 4*).

## Dysregulation of downstream regulators of cardiac ion channels

To gain insight into molecular mechanisms leading to differential susceptibility to developing diLQT, we performed a transcriptomic profiling of iPSC-CMs from high-S vs. low-S groups using RNA-sequencing. We looked for differentially expressed genes (DEGs) between the two groups (all cell lines included) and used a prior knowledge based approach to identify potential risk markers that could be mechanistically connected to QT prolongation development. By analyzing the LQTS neighborhood in the human interactome we recently demonstrated that graph theoretical models allow the prediction of new gene variants and drug targets that are involved in disease pathogenesis (*Berger et al., 2010*). Here, we used a similar approach and searched for differentially expressed direct neighbors of the known congenital LQTS genes in the human interactome (*Figure 5A*). Of note, expression of these LQTS genes was similar between high-S and low-S groups. We identified four up-regulated (*DLG2, KCNE4, PTRF,* and *HTR2C*) and one down-regulated direct neighbors (*CAMKV*) (*Figure 5A–B*). Among these newly identified candidates, *DLG2* (a member of the family of anchoring proteins called MAGUK), *KCNE4* (a regulatory sub-unit of potassium channel), *CAMKV* (a kinase-like protein that, in the presence of calcium, interacts with calmodulin (*CALM1*), and *PTRF* (Polymerase I and transcript release factor, or cavin-1) were relevant candidates as downstream regulators of cardiac ion channels.

The expression levels of the genes between the low-S and the high-S groups show that the risk specific expression (i.e. low expression for *CAMKV* and high expression for the other direct neighbors) does not accumulate on a few high-S subjects but is more or less equally distributed over all high-S subjects (*Figure 5B*). Each subject has a risk-associated expression of at least one direct neighbor, arguing for multiple ways to disturb the LQTS genes and therefore confer different susceptibility to develop diLQT.

Since most of the high-sensitivity subjects were females, while most of the low sensitivity subjects were males, we then estimated whether gender could impact our results. To neutralize gender-specific effects on gene expression, we generated new groups consisting of one high-S and one low-S subject of the same gender (16 female groups, six male groups, *Table 5*). DEGs were identified between all possible combinations of one female versus one male group (16 × 6 = 96 sets of DEGs). We counted for each gene how many sets of DEGs identified the gene as significantly up-regulated or down-regulated. Down-regulated counts were subtracted from up-regulated counts and resulting netto counts were normalized to the number of total comparisons (i.e. divided by 96). Using this approach, a gene that is up-regulated in the high-S group because of gender specific expression, should be identified as significantly up-regulated in most of the female versus male groups sets of DEGs. The same accounts for down-regulated genes. As shown in *Figure 5B and C*, this trend was not observed for most of the genes, arguing against gender differences being the reason for differential gene expression between the high-S and low-S group.

Finally, to further investigate if the DEGs are the consequence of sex specific gene expression, we predicted regulatory transcription factors of the DEGs based on two transcription factor target databases and ranked them by significance (*Figure 5—figure supplement 1*). The sex-hormone related transcription factors were ranked 16 (AR), 20 (ESR1) and 24 (ESR2) or 27 (PGR) and 180 (ESR1), and no sex hormone related transcription factor was among the top 10 candidates. These data suggest that the observed differences in expression patterns between high-S and low-S iPS-CMs are mostly indicative of a signature associated with drug sensitivity rather than a gender effect.

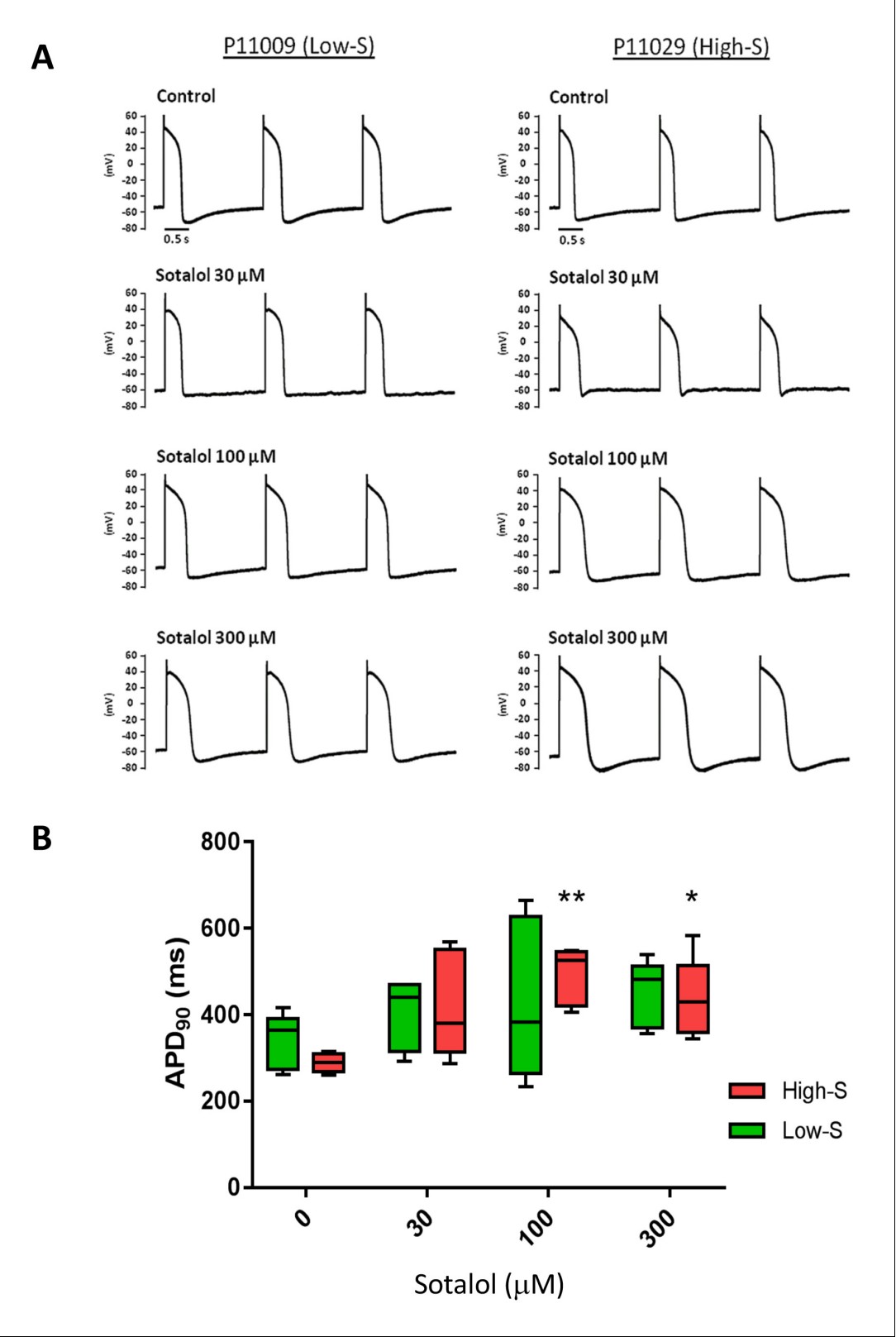

**Figure 4.** Patch-clamp analysis of action potential (AP) in representative iPSC-CMs. (**A**) Representative AP tracings of the iPSC-CMs generated from the low-S (P11009) and high-S (P11029) lines in control and sotalol-treated conditions. (**B**) bar chart summarizing the $APD_{90}$ in control and sotalol-treated conditions for the hiPSC-CMs
*Figure 4 continued on next page*

*Figure 4 continued*

generated from both low-S (P11009) and high-S (P11029) hiPSC cell lines (n = 5 for each condition). *p<0.05;
**p<0.01, ANOVA, followed by Tukey's, sotalol-treated versus respective control without sotalol application.

## Discussion

Our data illustrate the potential of using a genetically diverse panel of subject-specific iPSCs to model complex and acquired phenotypes. So far, iPSCs technology has been successful in recapitulating monogenic diseases with clear causative mutations including congenital long QT syndrome (*Itzhaki et al., 2011*; *Moretti et al., 2010*; *Sinnecker et al., 2014*). In contrast, diLQT develop in clinically normal individuals who have a genetic predisposition requiring an additional stressor to become manifest (*Roden, 2006*, *2008b*). The exact nature of this predisposing genetic background remains uncertain (*Petropoulou et al., 2014*) but is likely made of an allelic series of common variants with modest independent effect on cardiac repolarization but potential for a stronger impact in a polygenic model, as exemplified by our genetic profiling.

Here, we provide evidence that susceptibility to develop diLQT can be best reproduced in vitro using a targeted library of iPSCs derived from patients presenting with an extreme pharmacodynamic response to drug stimulation in vivo. Extreme phenotype selection is a well-defined methodology that helps enrich the frequency of alleles that contribute to a trait and this methodology has been successfully applied in genetic studies aimed at identifying variants associated with complex traits (*Cirulli and Goldstein, 2010*). The use of this methodology is particularly suitable when investigating variability in drug responses by segregating individuals with opposite predisposing genetic makeup, a characteristic that can be transmitted to iPSCs as shown by our data. Importantly, in order to avoid experimental biases, in vitro assessment of iPSC-CMs sensitivity to sotalol was performed blinded to the clinical phenotype. In addition, as we aimed at reproducing a more complex phenotype, we designed our study with a larger sample size than typical studies using iPSCs for disease modeling of monogenic disorders. As a consequence, we were only able to explore one iPSC clone per patient and we did not evaluate potential inter-line variability. The likelihood that inter-line variability would have systematically biased our results to reproduce the anticipated effect as seen in vivo and thus explain our results is however unlikely on a panel of 17 different cell lines as appreciated by low p-values for comparison between groups (*Figures 3* and *4*). Under our hypothesis, it is likely that the discrepant results obtained in two cell lines reflect that potential false results associated with inter-line variability from a same patient. However, external replication would further strengthen the potential of subject-specific iPSCs to model complex phenotypes as shown in our seminal study.

This library also represents a unique opportunity to explore the molecular mechanisms underlying pro-arrhythmic adverse drug reactions. The transcriptomic comparison of the generated iPSC-CMs

**Table 4.** Related to *Figure 4*. AP parameters of low-S and high-S iPSC-derived cardiomyocytes at baseline control condition. AP data are mean ± SE. APD50/APD90, AP duration measured at 50% or 90% repolarization; MDP, maximum diastolic potential. None of the baseline AP parameters was significantly different between the two cell lines.

| | Low-S (11009) (n=7) | High-S (11029) (n=5) |
|---|---|---|
| Firing Frequency (mV) | 1.26 ± 0.29 | 1.03 ± 0.28 |
| Amplitude (mV) | 88.3 ± 3.6 | 85.1 ± 4.7 |
| Upstroke velocity (mV/ms) | 11.7 ± 1.0 | 14.0 ± 2.0 |
| Decay velocity (mV/ms) | −7.2 ± 1.4 | −11.1 ± 3.5 |
| APD50 (ms) | 306.4 ± 57.9 | 299.4 ± 22.0 |
| APD90 (ms) | 341.8 ± 58.6 | 338.4 ± 21.4 |
| MDP (mV) | −69.4 ± 3.2 | −68.0 ± 1.3 |

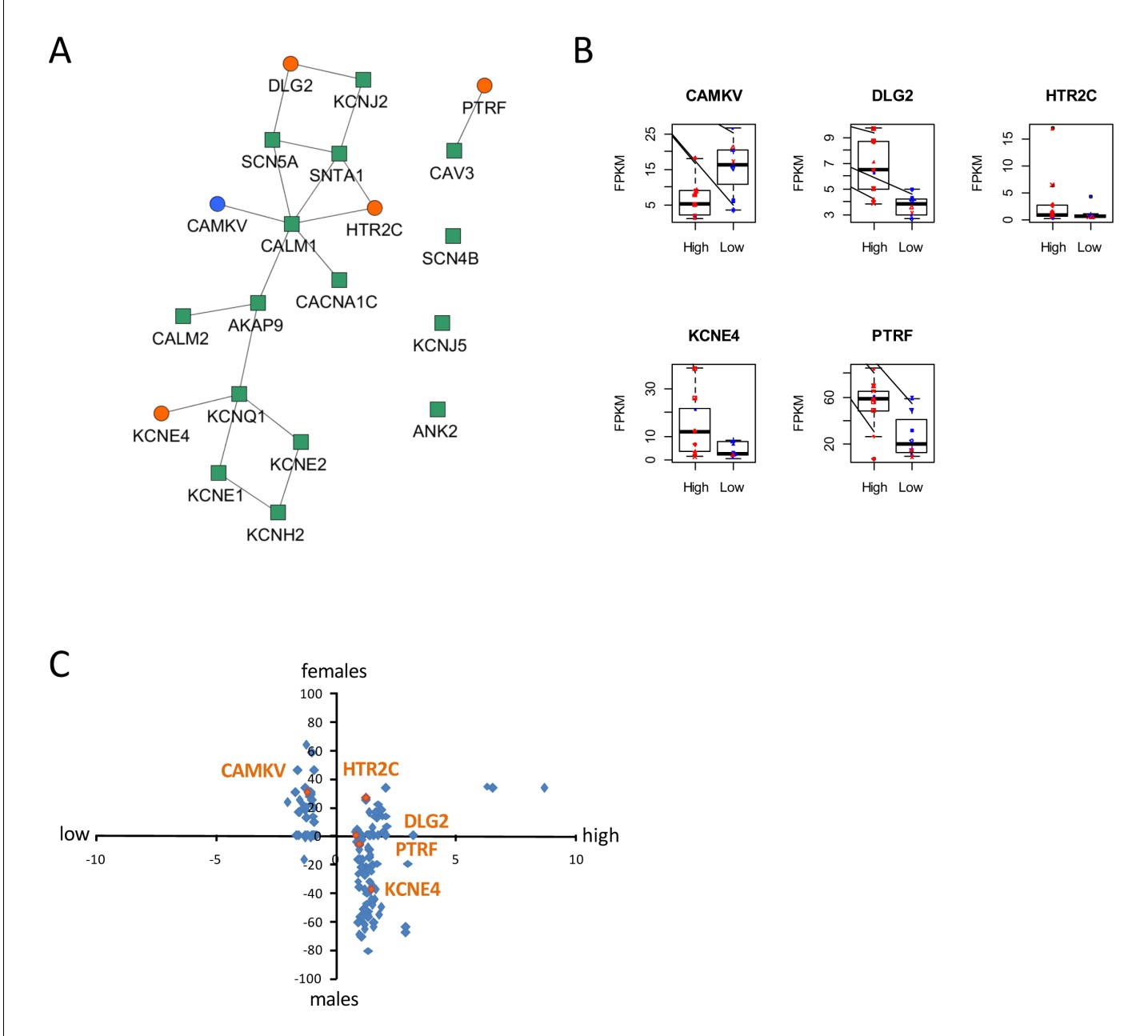

**Figure 5.** Identification of dysregulated genes as direct neighbors of QT-associated network in high-S iPSC-CMs. (**A**) Known LQTS genes were used as seed nodes (green squares) in the human interactome and five differentially expressed direct neighbors were identified (circles) (path length 1). Up-regulated genes are colored orange, down-regulated genes are in blue. (**B**) Relative expression of identified genes in each group. Males are represented in blue and females in red. Individual samples are represented by the same symbols in all diagrams. (**C**) Comparison of the log2-fold changes between the high-S and the low-S groups with the normalized counts of how often a gene (blue or orange dots) was found to be up- or down-regulated between the male and female groups. As there are more females in the high-S group and more males in the low-S groups, report of genes in the lower left or upper right quadrants indicate a gender-specific effect while the lower right and upper left quadrants argue for the lack of gender-specific effect. Except for *HTR2C*, dysregulation of all other candidate genes was suspected to occur independently of gender. There is one figure supplement.

The following figure supplement is available for figure 5:

**Figure supplement 1.** Prediction of sex hormones-related transcription factors.

**Table 5.** Related to *Figure 5*. Generation of new groups associating one high-S and one low-S line of the same gender (six groups for males and 16 groups for females) in order to neutralize gender-specific effects on gene expression. These groups were then used to determine DEGs between the high-S and low-S groups independent of gender.

| Male groups | Female groups |
| --- | --- |
| P11019 (high) + P11028 (low) | P11008 (high) + P11030 (low) |
| P11019 (high) + P11026 (low) | P11015 (high) + P11030 (low) |
| P11019 (high) + P11020 (low) | P11013 (high) + P11030 (low) |
| P11019 (high) + P11009 (low) | P11029 (high) + P11030 (low) |
| P11019 (high) + P11031 (low) | P11018 (high) + P11030 (low) |
| P11019 (high) + P11007 (low) | P11024 (high) + P11030 (low) |
| | P11023 (high) + P11030 (low) |
| | P11021 (high) + P11030 (low) |
| | P11008 (high) + P11014 (low) |
| | P11015 (high) + P11014 (low) |
| | P11013 (high) + P11014 (low) |
| | P11029 (high) + P11014 (low) |
| | P11018 (high) + P11014 (low) |
| | P11024 (high) + P11014 (low) |
| | P11023 (high) + P11014 (low) |
| | P11021 (high) + P11014 (low) |

from both groups combined with bioinformatics network analysis revealed significant changes in expression of genes related to the regulation of ion channels homeostasis (*DLG2, KCNE4, CAMKV* and *PTRF*). Of note, while none of these genes have been previously associated with the occurrence of diLQT, *CAMKV* is coding for a kinase-like protein that, in the presence of calcium, interacts with calmodulin (*CALM1*), a critical regulator of ion channels involved in congenital long QT syndrome 14 (*Crotti et al., 2013*) and in loperamide-induced long QT (*Berger et al., 2010*). Additionally, mutations in *PTRF* (Polymerase I and transcript release factor, or cavin-1) cause a particular form of congenital generalized lipodystrophy (type 4, CGL4) that is associated with features of long QT syndrome and high rate of sudden cardiac death (*Rajab et al., 2010*). *PTRF* is involved in the formation of caveolae (*Nabi, 2009*), that are critical in cardiac ion channels trafficking to the plasma membrane (*Maguy et al., 2006*). In addition to the involvement of *DLG2* and *KCNE4* genes in the direct regulation of scaffolding, trafficking and gating kinetics of some cardiac potassium channels (*Leonoudakis et al., 2004*; *Levy et al., 2010*), our study suggest for the first time that diLQT might be related to changes in downstream regulation of the cardiac repolarization machinery. Further experiments will be required to understand the exact contribution of each of these candidates and whether they might act in combination to confer susceptibility to develop diLQT. Similarly, future studies using single-cell RNA sequencing on cardiomyocytes derived from our library of subject-specific iPSCs would be needed to better identify and characterize differentially expressed genes and pathways. Lastly, we identified an up-regulation in *HTR2C* (5-hydroxytryptamine receptor 2C), a G-protein coupled receptor for serotonin that is expressed in the heart. Upon activation HTR2C activates down-stream signaling cascades to promote the release of Ca2+ from internal stores. Interestingly, serotonin receptor agonists have been associated with higher risk for diLQT (*Keller and Di Girolamo, 2010*; *Sarganas et al., 2014*). Intriguingly, we found non-significant trends for differences between groups in response to E-4031, a selective hERG channel blocker. Sotalol primarily prolongs cardiac repolarization through hERG inhibition but also have a non-selective competitive $\beta$-adrenergic receptor blocker activity and can thus affect other cardiac repolarization components that are associated with higher susceptibility to develop diLQT. Whether sotalol has a direct or indirect effect on the candidates identified in this study would deserve further investigations.

Many studies have reported a higher susceptibility to diLQT in premenopausal women as compared to men including in response to sotalol (*Darpo et al., 2014*; *Makkar et al., 1993*), an observation that is in line with the results of our prospective clinical investigation. It has been suggested that sex hormones can influence cardiac repolarization by modulating expression and function of cardiac ion channels but their effect is complex to appreciate as levels fluctuate with time and age and the different sex hormones can have counterbalancing effects (*Hulot et al., 2003*; *Odening and Koren, 2014*). Here, we were able to reproduce susceptibility to develop diLQT in vitro in a hormone-free environment. We performed specific analyses that did not support an important role of gender to explain our results, including the lack of a gender-specific gene expression signature and the lack of significant sex hormone-related transcription factors. Importantly, our data suggest that the identified genes should work as predisposing genes in both genders thus arguing for the existence of an intrinsic predisposing background that is carried by iPS cells but do not depend on gender. Of note, it has recently been shown that women have in mean a greater intrinsic sensitivity to sotalol as compared to men, an effect that however remains widely variable at the individual level (*Darpo et al., 2014*).

The development of a new generation of human-based screening assays for testing individual drug reactions (*Inoue and Yamanaka, 2011*; *Mann, 2015*) could help in better predicting adverse drug effect that usually develop in a relatively low proportion of susceptible individuals in the general patient population (*Budnitz et al., 2006*). Interestingly, we were able to detect significant differences in response to the lowest sotalol concentrations (i.e., 10 µM) in iPSC-derived cardiomyocytes from subjects clinically presenting with high sensitivity to sotalol. This suggests that the iPSC-derived platform could efficiently detect individual differences in the range of concentrations that is near the typical sotalol plasma levels. Finally, further developments in differentiation protocols (*Burridge et al., 2014*), in functional maturation of iPSC-derived cells (*Lundy et al., 2013*) or in iPSC-based bioengineering (*Schaaf et al., 2011*; *Turnbull et al., 2014*) might also improve the overall predictability of the method.

Our data illustrate the potential of iPSCs technology for the prediction of individual risk and for its use in precision medicine. We found a limited number of lines with discrepant results in vitro as compared to the clinical phenotype, suggesting a good accuracy of the approach. However, as we only investigated patients with extreme responses to sotalol, our study was not designed to accurately assess the predictive value of the test. This should be adequately addressed in larger prospective studies.

In conclusion, this study underscores the power of developing a panel of iPSCs to model complex traits such as susceptibility to develop cardiotoxic drug response.

## Material and methods

### Subjects and clinical investigations

To participate to the study, male or female volunteers had to fulfill the following inclusion criteria: aged 18 to 40 years, body mass index from 19 to 29 kg/m$^2$, no previous history of cardiac disease, no major disorders, no medications known to affect cardiac repolarization (as defined by https://www.crediblemeds.org/everyone/composite-list-all-qtdrugs). All volunteers had normal laboratory evaluations and a normal clinical examination with a resting heart rate $\geq$50 beats min$^{-1}$ and a systolic blood pressure $\geq$100 mmHg. Cardiac conduction and repolarization was assessed on a 12-lead resting electrocardiogram and participants were excluded in case of atrio-ventricular conduction disorder (PR interval >200 ms), QRS >100 ms and in case of prolonged QTcF >450 ms. Other exclusion criteria were familial or personal history of sudden death or unexplained syncope, concomitant use of illicit drugs, asthma.

The protocol was approved by the local institutional ethics committee and all subjects gave their written informed consent for participation to the study.

All investigations were performed prospectively at a single clinical center (Biotrial Paris, Rueil Malmaison, France). In a first phase, eligible subjects were admitted at 7:30am at the clinical investigation center after an overnight fast. Subjects were placed in a quiet room and rested for 30 min in the supine position. Cardiac rhythm and non-invasive blood pressure monitoring were started. The quality of ECG recordings was assessed and baseline recordings were performed during the 5 min

preceding sotalol oral administration (80 mg tablet). ECG recordings were then performed 3 hr after sotalol intake, i.e. around peak plasma concentration. Blood samples for the determination of sotalol concentration were obtained at baseline and 3 hr after sotalol administration. Volunteers were allowed to leave the clinical investigation center 5 hr after sotalol intake after cardiac repolarization had normalized.

In a second phase (7 to 35 days after phase 1), 20 subjects with the most extreme responses to sotalol (10 with smallest QTcF changes and 10 with the largest QTcF changes from baseline in response to sotalol) were asked to come to the clinical investigation center and had a skin biopsy under local anesthesia using a sterile 3 mm skin punch.

## ECG recording and analysis

Before and 3 hr following sotalol administration, 30 s digital 12-lead electrocardiograms were recorded using a Cardioplug device (Cardionics Inc, Brussels, Belgium) connected to a personal computer. All electrocardiogram recordings were then read by the same investigator, blindly of drug administration. The same chest lead with the largest T-wave amplitude was selected for QT interval measurements in a given subject. QT interval was measured manually directly on the computer screen by changing position of cursors indicating the start and the end of the cardiac interval: RR (interval between two successive R waves) and QT (tangent method). Baseline QTc was assessed as the mean of three electrocardiographic recordings obtained within 5 min before drug administration. QT was corrected by the Fridericia cubic root formula (QTcf), which minimizes the errors due to the square root Bazett formula. The clinical phenotype was defined by modifications in QTcf duration (ms) 3 hr after sotalol intake.

## Human iPSCs derivation

Fibroblast derivation was performed immediately after biopsy arrival at Ectycell / Cellectis Stem Cell Company (Evry, France). Skin biopsy of a 3 mm diameter punch was cut into about 20 small explants with scalpels and placed in 6-well plates and cultured with fibroblast growth medium containing DMEM medium (Gibco), 10% of FBS (PAA Laboratories), 10 ng/ml FGF-2 (Invitrogen, Carlsbad, CA) and 1% penicillin-streptomycin (Gibco). Cells were fed every 2–3 days. In this culture, fibroblasts appeared and became confluent on average 33 days after plating. The fibroblasts were then subcultured using 0.25% trypsin (Invitrogen) then re-suspended and cultured into T75 flasks using fibroblast growth medium without antibiotics. An average of 62 million fibroblasts per biopsy was obtained in 4–7 weeks at passage 1. The fibroblasts were tested for mycoplasma contamination and no contamination was detected (MycoAlert TM Mycoplasma Detection kit, Lonza France).

Reprogramming of fibroblasts derived from patient's skin biopsies into iPSCs was carried out using polycistronic retroviral vectors. The fibroblasts were seeded in 6-well plates at a defined cell density (100,000 cells/well) and cultured with fibroblast growth medium (DMEM medium containing 10% of FBS and 10 ng/ml FGF-2). For transduction, cells were incubated with polycistronic retroviral vectors (provided by Vectalys, Toulouse, France) carrying human Oct4, Sox2, Klf4, and c-Myc (OSKM) expression factors in fibroblast medium supplemented with 6 µg/ml polybrene (Sigma, St Louis, MO) overnight. The cells were transduced at a multiplicity of infection of 5. Four days post-infection, the cells were split by using 0.25% trypsin and plated at 100,000 cell/well in 0.1% gelatin-coated 6-well plates in fibroblast growth medium. After 24 hr, the medium was switched to the human pluripotent stem cell medium (DMEM/F12, 20% knockout serum replacement, 1× non-essential amino acid, 2 mM L-glutamine, 0.1 mM 2-mercaptoethanol, 20 ng/ml FGF-2 and 1% penicillin-streptomycin) with 0.5 mM Valproic acid (VPA). The media were changed every day. Around 20–25 days post-infection, the human ESC-like colonies appeared. The iPSCs colonies were picked and expanded in mTeSR1 (Stemcell technologies, Canada) on matrigel matrix (BD Biosciences France) coated plates. The iPSCs were passaged with Accutase (10 ml pre 75 cm2 surface area, Millipore, Billerica, MA) and the culture medium was changed daily.

Colonies fulfilling established 'stemness' criteria were selected and sent to the Cardiovascular Research Center at Mount Sinai School of Medicine, New York, USA for differentiation toward the cardiomyocyte lineage and pharmacological characterization.

## Differentiation of iPSCs into cardiomyocytes

The iPSCs were differentiated into cardiomyocytes using a directed differentiation method. Cardiomyocytes differentiation was initiated in suspension cultures on ultra-low attachment dishes (Corning, France) in mTESR1 medium supplemented with BMP4 10 ng ml-1) and Blebbistatin (5 μM) for 24 hr. The medium was then replaced with the basal differentiation medium (StemPro34, 50 μg ml-1 ascorbic acid, 2 mM GlutaMAX-I) supplemented with BMP4 (10 ng ml-1) and Activin-A (25 ng ml-1) for 48 hr (days 1–3) and then switched to basal differentiation medium for another 36 hr (days 3–4.5). Finally, the cells were differentiated in basal differentiation medium supplemented with IWR-1 (2.5 μM) for 96 hr (Day 4.5–8.5). The differentiated cardiomyocytes were maintained in basal differentiation media for up to four weeks. All cytokines were purchased from R and D. The small molecules were purchased from Sigma. All differentiation cultures were maintained in 5% $CO_2$/air environment.

## Immunocytochemistry

iPSCs were cultured on matrigel-coated coverslips, fixed in paraformaldehyde and permeabilized in blocking/permeabilization buffer (2% BSA/2% FBS/0.05% NP-40 in PBS) for 45 min and incubated with primary antibodies overnight at 4°C. Then the cells were washed in PBS and incubated with Alexa-conjugated secondary antibodies (Invitrogen) diluted in blocking/permeabilization buffer (1:750). Finally, after washing in PBS the cells were counterstained with DAPI. Immunofluorescence images were acquired using an Olympus X41 microscope. The following antibodies were used: mouse monoclonal anti-OCT4 (Santa Cruz biotechnology, Germany), goat polyclonal anti-NANOG (R and D systems), mouse monoclonal anti-SOX2 (R and D systems), mouse monoclonal anti-SSEA-4 (R and D systems, Minneapolis, MN), mouse monoclonal anti-TRA-1–60 (R and D systems). Similarly, iPSC-derived cardiomyocytes were dissociated and cultured on matrigel-coated coverslips for 4–5 days, fixed in paraformaldehyde and permeabilized in blocking/permeabilization buffer for 45 min. The following primary antibodies were used: mouse monoclonal anti-cardiac troponin T (Thermo Fisher Scientific, France), mouse monoclonal anti-connexin 43, and mouse monoclonal anti-α-actinin. Confocal imaging was performed using a Leica SP5 confocal system.

## Flow cytometry

Single-cell suspensions were obtained by dissociating EBs with 0.025% trypsin for 15 min at 37°C. The cells were then fixed with 4% paraformaldehyde for 15 min and washed twice with phosphate buffered saline (PBS). The fixed cells were first permeabilized in permeabilization buffer (0.2% Triton X-100 in PBS) for 30 min and then blocked with 10% goat serum for 25 min. Cells were then incubated with the primary antibody (anti-cardiac troponin T; Thermo Fisher Scientific). After 1 hr, the cells were washed in PBS, incubated with Goat anti-mouse IgG1 - Alexa 488 secondary antibody for 45 min, and finally washed twice with PBS. All procedures were performed at 4°C. Fluorescence-activated cell sorting analysis was carried out using a BD LSR analyzer (BD Biosciences).

## Quantitative PCR

Relative gene expression was determined using a two-steps quantitative real-time PCR method. Total RNA was isolated with the RNeasy Isolation kit with on-column DNase I treatment (Qiagen, Germany) and reverse-transcribed using the cDNA Synthesis Kit (Biorad, Hercules, CA). Quantitative RT-PCR was performed with the Quanta SYBR Green Supermix (Quanta biosciences, Beverly, MA) on the ABI Prism 7500 Real Time PCR System (Applied Biosystems, Foster City, CA). Fold changes in gene expression were determined using the comparative CT method (△△Ct) with normalization to the housekeeping gene B2M.

## SNPs genotyping array

Single nucleotide polymorphism (SNP) genotyping analysis was performed using the Illumina HumanOmni2.5–8 beadchip genotyping array, which comprise a comprehensive set of around 2.6 million SNPs (with MAF >2.5%) across the genome. The list of mapped SNPs can be found at http://support.illumina.com/downloads/infinium-omni2-5-8-v1-3-support-files.html. All genomic DNA was isolated from iPSC clones using Quick-gDNA Mini Prep kit (Zymo Research, Irvine, CA). Input genomic DNA (200 ng) was processed, hybridized to the array and scanned on an Illumina HiScan at the

Mount Sinai Genomic Core Facility. Based on chromosome coordinates, we extracted SNPs in *AKAP9, ANK2, CACNA1C, CALM1, CALM2, CAV3, KCNH2, KCNE1, KCNE2, KCNJ2, KCNJ5, KCNQ1, SCN4B, SCN5A, SNTA1* using Genome Studio (Illumina, San Diego, CA). Genotypes were estimated and compared between low-S and high-S groups. We used HapMap-CEU data to estimate the anticipated minor allelic frequency in European population.

## RNA-Sequencing

Total RNA was extracted using Zymo columns and 2 µg were used to generate a RNA-seq sequencing library. Poly-A selection and mRNA-SEQ library preparation were performed at the Mount Sinai Genomics Core Facility. Sequencing (50 bases, paired ends) was performed using an Illumina HiSeq2500. Annotated reads were obtained using STAR and HTSeq and normalized to full library size.

Sequencing reads were aligned to the human reference genome 'hg19' using Tophat 2.0.8 (*Trapnell et al., 2009*), samtools-0.1.7 and bowtie 2.1.0 (*Langmead and Salzberg, 2012*) and differentially expressed genes were identified with cufflinks 1.3.0 (*Trapnell et al., 2010*).

The total number of sequenced reads in a sample influences the likelihood to detect a lowly to moderately expressed gene, especially in case of lower read counts (*McIntyre et al., 2011*). In consequence, it could happen that fewer genes are detected in a sample with a lower read count than in a sample with a higher read count. Such an experimental artifact might distort normalization including total reads as well as upper quartile normalization, the two normalization options that are offered by cufflinks. Both normalization approaches only change the number of reads that are associated with a gene, but not the number of identified genes. Consequently, the same number of normalized reads might be distributed over a different number of genes in two different samples, causing the detection of an equally expressed gene in both samples as differentially expressed. To prevent such an experimental artifact we randomly removed reads from the sequenced samples until every sample had the same number of read counts. Reads were aligned to the human reference genome hg19 with Tophat using the ensemble GTF file as a gene annotation reference and the option 'no-novel-juncs'. Output BAM files were directly subjected to Cufflinks to identify differentially expressed genes, using the options 'multi-read-correct', 'upper-quartile-norm' and 'frag-bias-correct' against the hg19 genome. Differentially expressed genes were identified based on a FDR of 5% and a minimum fold change ($\log_2((FPKM_{condition1}+1)/(FPKM_{condition2} + 1)) >= \pm \log_2(1.3)$).

We determined differentially expressed genes between the high-S (P11008, P11013, P11015, P11018, P11019, P11021, P11023, P11024, P11029) and the low-S (P11007, P11009, P11014, P11020, P11026, P11028, P11030, P11031) cell lines. To compare gene expression values of the identified candidates across individual samples, we subjected all samples as individual conditions to the cufflinks analysis pipeline in the same way as described above. We used the FPKM values of the cufflinks results file as a measure for normalized gene expression.

To analyze, if differentially expressed genes between the high-S and low-S group are caused by gender differences between the two groups, we generated all possible combinations of one high-S female and one low-S female (16 groups) and one high-S male and one low-S male (six groups). We determined differentially expressed genes between all possible combinations of one female group versus one male group (16 × 6 sets of DEGs). For each gene we counted in how many sets of DEGs it was detected as significantly up-regulated or down-regulated, subtracted the counts for down-regulated detections from the counts for up-regulated detections and divided the resulting number by the counts for sets of DEGs (i.e. by 96).

## Network analysis and direct neighbors identification

To search for differentially expressed direct neighbors of LQTS disease genes, we generate a human interactome by merging all protein-protein interaction databases of the Expression2 kinases suite (*Chen et al., 2012*), except the 'Predicted PPI' database, and a recently published protein-protein interaction network (*Rolland et al., 2014*). Small letter symbols of the merged interactome were replaced by their human homologues, using the Mouse Genome Informatics mouse-human orthology database and the NCBI mouse-human homologene database. Finally, we removed all network nodes that were not official human gene symbols as reported in the NCBI geneInfo database. Differentially expressed genes in the direct neighborhood of LQTS disease genes ('AKAP9', 'ANK2',

'CACNA1C', 'CALM1', 'CALM2', 'CAV3', 'KCNE1', 'KCNE2', 'KCNH2', 'KCNJ2', 'KCNJ5', 'KCNQ1', 'SCN4B', 'SCN5A', 'SNTA1') were identified (path length = 1).

## Prediction of upstream regulatory transcription factors

All differentially expressed genes were subjected to transcription factor target enrichment analysis using the Chea-background and Transfac database (downloaded from EnrichR [*Chen et al., 2013*]), as described previously (*Karakikes et al., 2014*).

## Extracellular field potential recordings (MEA assay)

For MEA recordings, we used 6well-MEA arrays, which contain six independent culture chambers, separated by a macrolon ring (60–6 well MEA 200/30 iR-Ti-rcr, Multichannel Systems, Germany). Inside each well, there is a field of nine electrodes with an internal reference electrode (*Figure 2— figure supplement 2*). The 6well-MEA arrays were prepared by pipetting 5 µl fibronectin solution (100 µg/mL, BD Biosciences) and incubated at 37°C for at least 1 hr. iPSC-CMs were dissociated using 0.025% trypsin for 5 min at 37°C and seeded onto prepared MEA plates using 5 µl of cell suspension in StemPro34 medium and then incubated at 37°C/5% CO2. The day after the cells were covered with 200 µL of StemPro34 media. MEA recordings were performed once the cells started to beat again (5–7 days). Field potentials of spontaneously beating cardiomyocytes were recorded using a high-resolution Micro Electrode Array (MEA) recording system (MEA60 system, Multi Channel Systems, Reutlingen, Germany, http://www.multichannelsystems.com) at 37°C. The baseline steady state was achieved following an equilibration period of about 15 min in vehicle medium (StemPro34 media). The tested drugs (E4031 and Sotalol, Sigma) were then added directly to each well. Dose-response experiments were performed with the following sequence: drugs were diluted in 200 µL of StemPro34 media; increasing concentrations of drug were added to a well of the MEA array in a cumulative manner; the FPD recordings were started 3 min after application of a given concentration of a drug, a timing that was found optimal to achieve steady-state changes; FPD recordings were then performed for two additional minutes. Each drug was tested in at least three different wells. Raw MEA data were acquired using QT-Screen Lite and MC-Rack softwares (Multi Channel Systems, Reutlingen, Germany). The data were then exported on QT-Analyzer Software to analyze the field potential duration and the inter-beat interval (Multi Channel Systems, Reutlingen, Germany). Parameters were averaged on the 2 min recordings. Inter-beat interval (in ms) was used to calculate the instantaneous beat rate. To account for the dependency between repolarization rate and the beating rate, FPD was firstly adjusted using the popular Bazett's formula (adjustedFPD = rawFPD / RR1/2). We however found that, as expected, this formula was associated with an important over-correction thus supporting the need for an alternative adjustment formula according to Funck-Brentano et al. (*Funck-Brentano and Jaillon, 1993*) (*Figure 2—figure supplement 3*, panel A). We thus plotted the FPD / inter-beat-interval pairs recorded at various beating rates under baseline conditions. The following equation with the same dimension as a Bazett's formula: adjustedFPD = rawFPD / RR$\alpha$, where alpha is a regression parameter, was applied to raw data in order to have a new estimate of the correction factor and results in appropriate corrections of FPD (*Figure 2—figure supplement 3*, panel B).

Cells were considered as developing arrhythmias in response to sotalol as they presented ectopic beats or spontaneous beating rates became irregular.

## Patch clamp recordings

hiPSC-derived embryoid bodies were enzymatically dissociated into single cardiomyocytes and plated on matrigel-coated glass coverslips. The action potential of cardiomyocytes was assessed in the current-clamp mode using whole-cell patch-clamp technique with a HEKA EPC10 amplifier and Pulse/PulseFit software (HEKA, Germany). Pipette solution consisted of: 110 mM potassium aspartate, 20 mM KCl, 10 mM HEPES, 1 mM MgCl2, 0.1 mM ATP (disodium salt), 5 mM ATP (magnesium salt), 5 mM phosphocreatine (disodium salt) and 1 mM EGTA (pH 7.2). The external bath solution contained 140 mM NaCl, 5 mM KCl, 1 mM MgCl2, 10 mM D-glucose, 1 mM CaCl2 and 10 mM HEPES (pH 7.4). hiPSC-CMs were perfused with either control or sotalol-containing Tyrode's solution for 5 min. The whole-cell condition was then reached. A hyperpolarization current (−30 pA) was applied to silence the spontaneous automaticity. The AP recording was then performed with a train

of current pulse injection (1000 pA, 5 ms) at 0.5 Hz steady pacing. The assay was performed at 37°C and finished within 30 min upon the drug perfusion.

## Statistical analysis

All analyses were performed using Prism 6.0 (GraphPad, La Jolla, CA). Continuous data are presented as Mean ± SEM. A p-value≤0.05 was considered significant. Continuous variables were compared using non-parametric Mann-Whitney test and binary variables were tested using Chi-square or exact Fisher's test as appropriate. For the in vivo investigations, we planned to identify 20 extreme responders to sotalol (i.e., 10 with high response defined by a deltaQTcf >35 ms and 10 with low response defined by a deltaQTcf <5 ms, therefore defining a minimal difference of 30 ms between groups). We then anticipated that this in vivo difference will be reproduced in vitro with a reduction by 33%. We thus determined that this sample size will give 80% power to demonstrate a difference of 20 ms between groups, with a standard error of 15 ms and an alpha-error risk of 0.05 (nQuery Advisor version 4.0).

Responses to drugs, as measured in vitro, were normalized to baseline values and expressed as changes to baseline to account for inter-line variability. All recorded data were analyzed. A two-way analysis of variance with repeated measures was performed to analyze response to sotalol. Sotalol concentrations define the first factor and the sensitivity group the second factor. Individual comparisons were performed when the overall analysis was significant. Dose response curves were built to estimate EC50 using the Hill equation.

## Acknowledgements

We thank the Biotrial core lab. This work was funded by the Cellectis Company. JH and RI are supported by NIH P50 GM071558 and GM54508 and U54 HG008098.

# Additional information

### Competing interests

SR, MW, YV, CD: Current or former employee of the Cellectis Company. The other authors declare that no competing interests exist.

### Funding

| Funder | Grant reference number | Author |
| --- | --- | --- |
| National Institutes of Health | P50 GM071558 | Jens Hansen<br>Ravi Iyengar |
| National Institutes of Health | GM54508 | Jens Hansen<br>Ravi Iyengar |
| National Institutes of Health | U54 HG008098 | Jens Hansen<br>Ravi Iyengar |
| Cellectis Company | Research subvention | Jean-Sébastien Hulot |

The funders had no role in study design, data collection and interpretation, or the decision to submit the work for publication.

### Author contributions

FS, RL, RI, Acquisition of data, Analysis and interpretation of data, Drafting or revising the article; JH, CF-B, J-ES, Analysis and interpretation of data, Drafting or revising the article; C-WK, LG, Acquisition of data, Analysis and interpretation of data; IK, SS, SR, MW, DJ, NZ, Acquisition of data, Drafting or revising the article; YV, Acquisition of data, Contributed unpublished essential data or reagents; CD, Conception and design, Contributed unpublished essential data or reagents; RJH, Conception and design, Analysis and interpretation of data, Drafting or revising the article; J-SH, Conception and design, Acquisition of data, Analysis and interpretation of data, Drafting or revising the article

## Author ORCIDs

Jean-Sébastien Hulot, http://orcid.org/0000-0001-5463-6117

## Ethics

Clinical trial registration clinicaltrials.gov NCT01338441

Human subjects: The protocol was approved by local institutional ethics committee (CPP Ile de France XI 11-015 & Afssaps A110094-37) and all subjects gave their written informed consent for participation to the study.

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
