## [Decision Letter]

Thank you for submitting your article "Modeling susceptibility to drug-induced long QT with a panel of subject-specific iPS cells" for consideration by *eLife*. Your article has been reviewed by three peer reviewers, and the evaluation has been overseen by Janet Rossant as the Senior Editor and Reviewing Editor. The reviewers have opted to remain anonymous.

The reviewers have discussed the reviews with one another and the Reviewing Editor has drafted this decision to help you prepare a revised submission.

Summary:

Stillitano and colleagues present an interesting translational study examining the ability of patient-specific iPS cells to report patient-specific responses to drug-induced long QT syndrome. This cardiotoxicity has led to multiple drug withdrawals as well as mandated safety testing for all new compounds seeking FDA approval. The authors have performed a clinical study in which they exposed volunteers to sotalol and measured changes in the electrocardiographic QT interval. They found that the patients exhibit a wide range of QTc responses. They then studied the extremes – high sensitivity and low sensitivity to sotalol. They generated patient specific iPS cell lines from the low and high sensitivity patients and differentiated them to cardiomyocytes, whose electrical properties were studied using a microelectrode array system. The authors found evidence that the adjusted field potential duration (aFDP), an in vitro surrogate for the QTc interval in the ECG segregated with the low and high sensitivity samples as assessed clinically. These findings support the hypothesis that iPSC-CMs can be used to report patient-specific responses to QT prolonging drugs. In addition, the authors did genomic analysis comparing high and low sensitivity samples finding some variability in SNPs. In addition, transcriptome analysis identified changes in expression levels of genes associated with cardiac repolarization comparing high and low sensitivity samples which is intriguing. The manuscript has several strengths. This is the first attempt to do such a prospective clinical trial rigorously testing iPSC-CM technology, and it addresses problem that is highly relevant for drug development, unanticipated drug-induced QT prolongation. The number of lines tested and characterized is substantial given the extent of analysis involved.

Essential revisions:

While the reviewers were impressed with the number of volunteers analyzed and the depth of the analysis, they had several concerns that raised questions about the validity of the results. Most importantly, they were concerned that the measurements of QT intervals were not undertaken using the most accurate and commonly accepted methodology, based on the example in Figure 1. For the high sensitivity example, the before measurement excluded the U wave and the after measurement includes the U wave thus extending the measured QT interval inappropriately in the post-sotalol measurement. Although there is not uniform consensus on QT interval measurement approaches, choosing the precordial wave with the largest T wave amplitude and variably including U waves is a potential problem that shakes confidence in the subsequent study that is absolutely dependent on the accuracy of these measurements. These measurements need to be substantiated.

Secondly, there was some concern that variability in cardiac differentiation efficiency among iPS lines could have influenced results. What measurements of iPSC-CM quality and purity were made routinely given that differentiation of different iPSC lines can yield different purity of CMs? This is a critical question since techniques taking whole preparations such as gene expression studies could be substantially impacted by contaminating cell types, e.g. 10% CMs from one line vs. 90% CMs from another line would not be expected to allow easy comparison of CM gene expression. This question is particularly raised because of the remarkable variability in various cardiac ion channel genes seen in Figure 2—figure supplement 3, with some lines show 10-20 fold differences in expression.

Third, the authors have found no differential response in the cells to E4031 (pure hERG blocker). As the primary effect of sotalol is, as stated by the authors themselves, hERG inhibition, why did their correlation between QTc and drug effects not work for E4031 as well? This needs to be discussed.

---

## [Author Response]

Essential revisions:

While the reviewers were impressed with the number of volunteers analyzed and the depth of the analysis, they had several concerns that raised questions about the validity of the results. Most importantly, they were concerned that the measurements of QT intervals were not undertaken using the most accurate and commonly accepted methodology, based on the example in Figure 1. For the high sensitivity example, the before measurement excluded the U wave and the after measurement includes the U wave thus extending the measured QT interval inappropriately in the post-sotalol measurement. Although there is not uniform consensus on QT interval measurement approaches, choosing the precordial wave with the largest T wave amplitude and variably including U waves is a potential problem that shakes confidence in the subsequent study that is absolutely dependent on the accuracy of these measurements. These measurements need to be substantiated.

We fully agree with this comment and we apologize as Figure 1 was indeed not submitted in its appropriate version. Figure 1 which was initially provided was mistakenly taken from a working version of the figure. We apologize for this inattention. The revised Figure 1 now shows typical ECG recordings in a high-sensitive and a low-sensitive subject. As detailed in the Methods section, we were indeed extremely careful in using highly accurate and standardized measurements of QT interval. A specific operating procedure was developed for ECG recordings, collections, transfer and analyses. The three consecutive complexes used for the measurements of the intervals were selected in periods with stable heart rate, stable baseline, absence of artifacts and U waves either absent or well differentiated from T-waves. Measurements were then performed using the tangent method by an investigator blindly of drug administration. Importantly, all ECG and measurements were reviewed by another cardiologist (still blinded of drug administration). As an extended example, we here provide the complete ECG recording and caliper positions (raw data, low resolution) for the high-s subject (baseline evaluation). All other ECG were recorded and measured in the same conditions (available on request).

The Methods section “ECG recording and analysis” has been updated accordingly.

Author response image 1.**DOI:**
http://dx.doi.org/10.7554/eLife.19406.021

Secondly, there was some concern that variability in cardiac differentiation efficiency among iPS lines could have influenced results. What measurements of iPSC-CM quality and purity were made routinely given that differentiation of different iPSC lines can yield different purity of CMs? This is a critical question since techniques taking whole preparations such as gene expression studies could be substantially impacted by contaminating cell types, e.g. 10% CMs from one line vs. 90% CMs from another line would not be expected to allow easy comparison of CM gene expression. This question is particularly raised because of the remarkable variability in various cardiac ion channel genes seen in Figure 2—figure supplement 3, with some lines show 10-20 fold differences in expression.

We agree that inter-line variability in the efficiency of differentiation has been reported. Therefore, we systematically performed cardiac troponin T (cTNT^+^) Flow Cytometry in all iPSC-CM lines at day 25 of differentiation. We found an average of 47.8±19.9% cTNT+ cells. Importantly, after unblinding, we did not observe any differences between iPSC lines from the high-S vs. low-S groups (49.2±24.4% vs. 46.7±17.3% respectively, p=0.42). This has been added to the Results section (subsection “Derivation and differentiation of iPSCs into cardiomyocytes”, last paragraph).

For measurements of iPSC-CMs quality, each line was tested for the presence of key CMs identity markers (sarcomeric proteins troponin T (cTnT), α-actinin and for the gap-junction protein connexin 43 (Cx43)) by immunostaining. A representative image is reported in Figure 2. The MEA recordings were performed on a layer of iPSC-CMs that had spontaneous beating activity and had measurable field potential duration (FPD) as they developed contact with the electrodes. It seems unlikely that contaminating cell types would have influenced these results. In addition, patch clamp experiments were performed at the cell level on two different couples of iPSC-CM lines from each group (low-S and high-S) and confirmed the MEA results (see previous and new Figure 4). Finally, our gene expression studies were indeed performed on whole preparations because of challenges in enriching preparations in derived cardiomyocytes. For our transcriptomic study, we thus decided to use a prior knowledge based approach (on cardiac repolarization / LQTS neighborhood; based on a previously published methodology) to identify signals that are specifically coming from cardiomyocytes in these preparations. Indeed, a direct neighbor of a LQTS gene is anticipated to be cardiomyocyte specific. The expression of cardiac ion channel genes indeed appears variable (Figure 2—figure supplement 3) but was similar between both groups (Low-S and high-S), as also further confirmed in our transcriptomic analysis (Figure 5 and Results section, subsection “Dysregulation of downstream regulators of cardiac ion channels”). For all of these reasons, we think we have limited the influence of variability in cardiac differentiation efficiency among iPS lines on our results. Our multiple investigations did not indicate a systematic bias favoring one group as compared to the other.

Third, the authors have found no differential response in the cells to E4031 (pure hERG blocker). As the primary effect of sotalol is, as stated by the authors themselves, hERG inhibition, why did their correlation between QTc and drug effects not work for E4031 as well? This needs to be discussed.

We thank the reviewers and editors for this comment. As pointed out, E-4031 is an experimental drug that is a selective hERG channel blocker whereas sotalol is a clinically used drug that is a non-selective competitive β-adrenergic receptor blocker with additional Class III antiarrhythmic properties by its inhibition of potassium channels. We found that trends in response to E-4031 (both for EC50 and maximal effect) were in the same direction than with sotalol but these trends did not reach statistical significance. The two compounds differ in their pharmacological characteristics in terms of inhibition potency and target selectivity. First, as compared to sotalol, E-4031 is a more potent hERG blocker with an IC_50_ value reported to be in the range of 10-100nM (vs. 10-50 µM for sotalol) and presents a very steep concentration-response effect (as observed in our lines, Figure 2—figure supplement 4C). It is thus possible that subtle differences in the susceptibility to develop prolonged cardiac repolarization in response to a given drug cannot be fully revealed when using a very potent inhibitor such as E-4031. Second, sotalol is thought to primarily prolong cardiac repolarization through hERG inhibition but sotalol can also affect other cardiac repolarization components. Our own transcriptomic results indicate that higher susceptibility to diLQT was associated with significant changes in expression of genes related to the regulation of ion channels homeostasis while the expression of cardiac ion channels (including hERG) was similar between both groups. A direct or indirect effect of sotalol on these components (in addition to its hERG effect) could explain the apparently discrepant results obtained with E-4031. This hypothesis would however deserve further investigations. The discussion has been corrected accordingly.